# ImageNet-21K Pretraining for the Masses

**Tal Ridnik**
DAMO Academy, Alibaba Group
tal.ridnik@alibaba-inc.com

**Emanuel Ben-Baruch**
DAMO Academy, Alibaba Group
emanuel.benbaruch@alibaba-inc.com

**Asaf Noy**
DAMO Academy, Alibaba Group
asaf.noy@alibaba-inc.com

**Lihi Zelnik-Manor**
DAMO Academy, Alibaba Group
lihi.zelnik@alibaba-inc.com

## Abstract

ImageNet-1K serves as the primary dataset for pretraining deep learning models for computer vision tasks. ImageNet-21K dataset, which is bigger and more diverse, is used less frequently for pretraining, mainly due to its complexity, low accessibility, and underestimation of its added value. This paper aims to close this gap, and make high-quality efficient pretraining on ImageNet-21K available for everyone. Via a dedicated preprocessing stage, utilization of WordNet hierarchical structure, and a novel training scheme called semantic softmax, we show that various models significantly benefit from ImageNet-21K pretraining on numerous datasets and tasks, including small mobile-oriented models. We also show that we outperform previous ImageNet-21K pretraining schemes for prominent new models like ViT and Mixer. Our proposed pretraining pipeline is efficient, accessible, and leads to SoTA reproducible results, from a publicly available dataset. The training code and pretrained models are available at: https://github.com/Alibaba-MIIL/ImageNet21K

## 1 Introduction

ImageNet-1K dataset, introduced for the ILSVRC2012 visual recognition challenge [45], has been at the center of modern advances in deep learning [30, 20, 46]. ImageNet-1K serves as the main dataset for pretraining of models for computer-vision transfer learning [51, 33, 21], and improving performances on ImageNet-1K is often seen as a litmus test for general applicability on downstream tasks [28, 62, 44]. ImageNet-1K is a subset of the full ImageNet dataset [11], which consists of 14,197,122 images, divided into 21,841 classes. We shall refer to the full dataset as ImageNet-21K, following [27] (although other papers sometimes described it as ImageNet-22K [8]). ImageNet-1K was created by selecting a subset of 1.2M images from ImageNet-21K, that belong to 1000 mutually exclusive classes.

Even though some previous works showed that pretraining on ImageNet-21K could provide better downstream results for large models [27, 14], pretraining on ImageNet-1K remained far more popular. A main reason for this discrepancy is that ImageNet-21K labels are not mutually exclusive - the labels are taken from WordNet [38], where each image is labeled with one label only, not necessarily at the highest possible hierarchy of WordNet semantic tree. For example, ImageNet-21K dataset contains the labels "chair" and "furniture". A picture, with an actual chair, can sometimes be labeled as "chair", but sometimes be labeled as the semantic parent of "chair", "furniture". This kind of tagging methodology complicates the training process, and makes evaluating models on ImageNet-21K less accurate. Other challenges of ImageNet-21K dataset are the lack of official train-validation split, the fact that training is longer than ImageNet-1K and requires highly efficient training schemes, and that the raw dataset is large - 1.3TB.

35th Conference on Neural Information Processing Systems (NeurIPS 2021) Track on Datasets and Benchmarks.

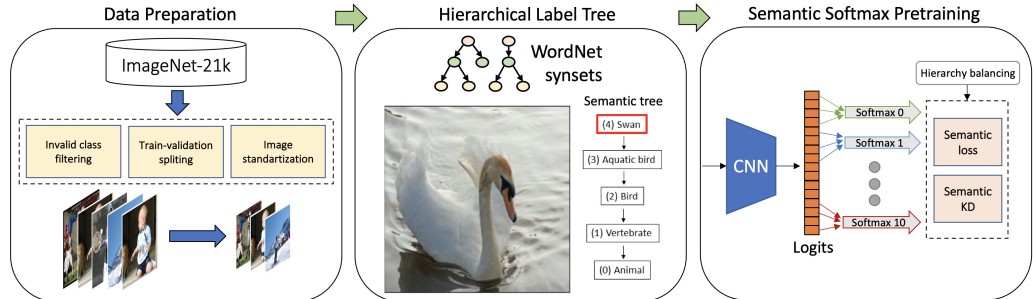

**Figure 1: Our end-to-end pretraining pipeline on ImageNet-21K.** We start with a dataset preparation and preprocessing stage. Via WordNet's synsets, we convert all the single-label inputs to semantic multi-labels, resulting in a semantic structure for ImageNet-21K, with 11 possible hierarchies. For each hierarchy, we apply a dedicated softmax activation, and aggregate the losses with hierarchy balancing.

Several past works have used ImageNet-21K for pretraining, mostly in comparison to larger datasets, which are not publicly available, such as JFT-300M [49]. [40] and [43] used ImageNet-21K and JFT-300M to train expert models according to the datasets hierarchies, and combined them to ensembles on downstream tasks; [27] and [14] compared pretraining JFT-300M to ImageNet-21K on large models such as ViT and ResNet-50x4. Many papers used these pretrained models for downstream tasks (e.g., [63, 41, 36, 1]). There are also works on ImageNet-21K that did not focus on pretraining: [61] used extra (unlabled) data from ImageNet-21K to improve knowledge-distillation training on ImageNet-1K; [13] used ImageNet-21k for testing few-shot learning; [56] tested efficient softmax schemes on ImageNet-21k; [17] tested pooling operations schemes on animal-oriented subset of ImageNet-21k.

However, previous works have not methodologically studied and optimized a pretraining process specifically for ImageNet-21K. Since this is a large-scale, high-quality, publicly available dataset, this kind of study can be highly beneficial to the community. We wish to close this gap in this work, and make efficient top-quality pretraining on ImageNet-21K accessible to all deep learning practitioners.

Our pretraining pipeline starts by preprocessing ImageNet-21K to ensure all classes have enough images for a meaningful learning, splitting the dataset to a standardized train-validation split, and resizing all images to reduce memory footprint. Using WordNet semantic tree [38], we show that ImageNet-21K can be transformed into a (semantic) multi-label dataset. We thoroughly analyze the advantages and disadvantages of single-label and multi-label training. Extensive tests on downstream tasks show that multi-label pretraining does not improve results on downstream tasks, despite having more information per image. To effectively utilize the semantic data, we develop a novel training method, called *semantic softmax*, which exploits the hierarchical structure of ImageNet-21K tagging to train the network over several semantic softmax layers, instead of the single layer. Using semantic softmax pretraining, we consistently outperform both single-label and multi-label pretraining on downstream tasks. By integrating semantic softmax into a dedicated semantic knowledge distillation loss, we further improved results. The complete end-to-end pretraining pipeline appears in Figure 1.

Using semantic softmax pretraining on ImageNet-21K we achieve significant improvement on numerous downstream tasks, compared to standard ImageNet-1K pretraining. Unlike previous works, which focused on pretraining of large models only [27], we show that ImageNet-21K pretraining benefits a wide variety of models, from larger models like TResNet-L [44], through medium-sized models like ResNet50 [20], and even small mobile-dedicated models like OFA-595 [5] and MobileNetV3 [21]. Our proposed pretraining scheme also outperforms previous ImageNet-21K pretraining schemes that were used to trained MLP-based models like Vision-Transformer (ViT) [14] and Mixer [53].

The paper's contribution can be summarized as follows:

- We develop a methodical preprocess procedure to transform raw ImageNet-21K into a viable dataset for efficient, high-quality pretraining.
- Using WordNet semantic tree, we convert each (single) label to semantic multi labels, and compare the pretrain quality of two baseline methods: single-label and multi-label pretraining. We show that while a multi-label approach provides more information per image, it can have significant optimization drawbacks, resulting in inferior results on downstream tasks.

- We develop a novel training scheme called semantic softmax, which exploits the hierarchical structure of ImageNet-21K. With semantic softmax pretraining, we outperform both single-label and multi-label pretraining on downstream tasks. We further improve results by integrating semantic softmax into a dedicated semantic knowledge distillation scheme.
- Via extensive experimentations, we show that compared to ImageNet-1K pretraining, ImageNet-21K pretraining significantly improves downstream results for a wide variety of architectures, include mobile-oriented ones. In addition, our ImageNet-21K pretraining scheme consistently outperforms previous ImageNet-21K pretraining schemes for prominent new models like ViT and Mixer.

## 2 Dataset Preparation

### 2.1 Preprocessing ImageNet-21K

Our preprocessing stage consists of three steps, as described in Figure 1 (leftmost image): (1) invalid classes cleaning, (2) creating a validation set, (3) image resizing. Details are as follows:
**Step 1 - cleaning invalid classes:** the original ImageNet-21K dataset [11] consists of 14,197,122 images, each tagged in a single-label fashion by one of 21,841 possible classes. The dataset has no official train-validation split, and the classes are not well-balanced - some classes contain only 1-10 samples, while others contain thousands of samples. Classes with few samples cannot be learned efficiently, and may hinder the entire training process and hurt the pretrain quality [23]. Hence we start our preprocessing stage by removing infrequent classes, with less than 500 labels. After this stage, the dataset contains 12,358,688 images from 11,221 classes. Notice that the cleaning process reduced the number of total classes by half, but removed only 13% of the original pictures.
**Step 2 - validation split:** we allocate 50 images per class for a standardized validation split, that can be used for future benchmarks and comparisons.
**Step 3 - image resizing:** ImageNet-1K training usually uses *crop-resizing* [22] which favours loading the original images at full resolution and resizing them on-the-fly. To make ImageNet-21K dataset more accessible and accelerate training, we resized during the preprocessing stage all the images to 224 resolution (equivalent to *squish-resizing* [22]). While somewhat limiting scale augmentations, this stage significantly reduces the dataset's memory footprint, from 1.3TB to 250GB, and makes loading the data during training faster.

After finishing the preprocessing stage, we kept only valid classes, produced a standardized train-validation split, and significantly reduced the dataset size. We shall name this processed dataset **ImageNet-21K-P** (P for Processed).

### 2.2 Utilizing Semantic Data

We now wish to analyze the semantic structure of ImageNet-21K-P dataset. This structure will enable us to better understand ImageNet-21K-P tagging methodology, and employ and compare different pretraining schemes.

**From single labels to semantic multi labels** Each image in the original ImageNet-21K dataset was labeled with a single label, that belongs to WordNet synset [38]. Using the WordNet synset hyponym (subtype) and hypernym (supertype) relations, we can obtain for each class its parent class, if exists, and a list of child classes, if exists. When applying the parenthood relation recursively, we can build a semantic tree, that enables us to transform ImageNet-21K-P dataset into a multi-label dataset, where each image is associated with several labels - the original label, and also its parent class, parent-of-parent class, and so on. Example is given in Figure 1 (middle image) - the original image was labeled as 'swan', but by utilizing the semantic tree, we can produce a list of semantic labels for the image - 'animal, vertebrate, bird, aquatic bird, swan'. Notice that the labels are sorted by hierarchy: 'animal' label belongs to hierarchy 0, while 'swan' label belongs to hierarchy 4. A label from hierarchy $k$ has $k$ ancestors.

**Understanding the inconsistent tagging methodology** The semantic structure of ImageNet-21K enables us to understand its tagging methodology better. According to the stated tagging methodology of ImageNet-21K [11], we are not guaranteed that each image was labeled at the highest possible hierarchy. An example is given in Figure 2. Two pictures, that contain the animal cow, were labeled differently - one with the label 'animal', the other with the label 'cow'. Notice that 'animal' is a

Ground-truth: **Animal**    Ground-truth: **Cow**

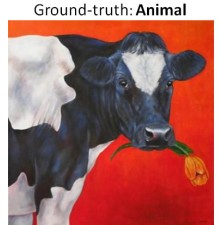 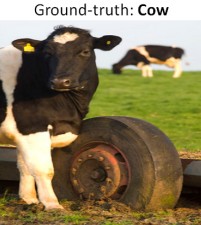

**Figure 2: Example of inconsistent tagging in ImageNet-21K dataset.** Two pictures containing the same animal were labeled differently.

| Hierarchy | Example Classes |
|-----------|-----------------|
| 0 | person, animal, plant, food, artifact |
| 1 | domestic animal, basketball court, clothing |
| ... | |
| 6 | whitetip shark, ortolan, grey kingbird |

**Table 1: Examples of classes from different ImageNet-21K-P hierarchies.**

semantic ancestor of 'cow' (cow → placental → mammal → vertebrate → animal). This kind of incomplete tagging methodology, which is common in large datasets [32, 42], hinders and complicates the training process. A dedicated scheme that tackles this tagging methodology will be presented in section 3.3.

**Semantic statistics**   By using WordNet synsets, we can calculate for each class the number of ancestors it has - its hierarchy. In total, our processed dataset, ImageNet-21K-P, has 11 possible hierarchies. Example of classes from different hierarchies appears in Table 1. In Figure 4 in appendix A we present the number of classes per hierarchy. We see that while there are 11 possible hierarchies, the vast majority of classes belong to the lower hierarchies.

## 3   Pretraining Schemes

In this section, we will review and analyze two baseline schemes for pretraining on ImageNet-21K-P: single-label and multi-label training. We will also develop a novel new scheme for pretraining on ImageNet-21K-P, *semantic softmax*, and analyze its advantages over the baseline schemes.

### 3.1   Single-label Training Scheme

The straightforward way to pretrain on ImageNet-21K-P is to use the original (single) labels, apply softmax on the output logits, and use cross-entropy loss. Our single-label training scheme is similar to common efficient training schemes on ImageNet-1K [44], with minor adaptations to better handle the inconsistent tagging (Full training details appear in appendix B.1). Since we aim for an efficient scheme with maximal throughput, we don't incorporate any tricks that might significantly increase training times. To further shorten the training times, we propose to initialize the models from standard ImageNet-1K training, and train on ImageNet-21K-P for 80 epochs. On 8xV100 NVIDIA GPU machine, mixed-precision training takes 40 minutes per epoch for ResNet50 and TResNet-M architectures ($\sim 5000 \frac{img}{sec}$), leading to a total training time of 54 hours. Similar accuracies are obtained when doing random initialization, but training the models longer - 140 epochs.

**Pros of using single-label training**

- **Well-balanced dataset** - with single-label training on ImageNet-21K-P, the dataset is well-balanced, meaning each class appears, roughly, the same number of times.
- **Single-loss training** - training with a softmax (a single loss) makes convergence easy and efficient, and avoids many optimization problems associated with multi-loss learning, such as different gradient magnitudes and gradient interference [60, 7, 9].

**Cons of using single-label training**

- **Inconsistent tagging** - due to the tagging methodology of ImageNet-21K-P, where we are not guaranteed that an image was labeled at the highest possible hierarchy, ground-truth labels are inherently inconsistent. Pictures, containing the same object, can appear with different single-label tagging (see Figure 2 for example).
- **No semantic data** - during training, we are not presenting semantic data via the single-label ground-truth.

## 3.2 Multi-label Training Scheme

Using the semantic tree, we can convert any (single) label to semantic multi labels, and train our models on ImageNet-21K-P in a multi-label fashion, expecting that the additional semantic information per image will improve the pretrain quality. As commonly done in multi-label classification [3], we reduce the problem to a series of binary classification tasks. Given $N$ labels, the base network outputs one logit per label, $z_n$, and each logit is independently activated by a sigmoid function $\sigma(z_n)$. Let's denote $y_n$ as the ground-truth for class $n$. The total classification loss, $L_{\text{tot}}$, is obtained by aggregating a binary loss from the $N$ labels:

$$L_{\text{tot}} = \sum_{n=1}^{N} L\left(\sigma(z_n), y_n\right). \tag{1}$$

Eq. 1 formalizes multi-label classification as a multi-task problem. Since we have a large number of classes $(11,221)$, this is an extreme multi-task case. For training, we adopted the high-quality training scheme described in [3], that provided state-of-the-art results on large-scale multi-label datasets such as Open Images [32]. Full training details appear in appendix B.2.

**Pros of using multi-label training**

- **More information per image** - we present for each image all the available semantic labels.
- **Tagging and metrics are more accurate** - if an image was originally given a single label at hierarchy k, with multi-label training we are guaranteed that all ground-truth labels at hierarchies 0 to k are accurate. Hence, multi-label training partly mitigates the inconsistent tagging problem, and makes training metrics more accurate and reflective than single-label training.

**Cons of using multi-label training**

- **Extreme multi-tasking** - with multi-label training, each class is learned separately (sigmoids instead of softmax). This extreme multi-task learning makes the optimization process harder and less efficient, and may cause convergences to a local minimum [60, 7, 15].
- **Extreme imbalancing** - as a multi-label dataset with many classes, ImageNet-21K-P suffers from a large positive-negative imbalance [3]. In addition, due to the semantic structure, multi-label training is hindered by a large class imbalance [24] - on average, classes from a lower hierarchy will appear far more frequent than classes from a higher hierarchy.

In appendices C.2 and E we show that for multi-label training, ASL loss [3], that was designed to cope with large positive-negative imbalancing, significantly outperforms cross-entropy loss, both on upstream and downstream tasks. This supports our analysis of extreme imbalancing as a major optimization challenge of multi-label training. Notice that we also list extreme multi-tasking as another optimization pitfall of multi-label training, and a dedicated scheme for dealing with it might further improve results. However, most methods that tackle multi-task learning, such as GradNorm [7] and PCGrad [60], require computation of gradients for each class separately. This is computationally infeasible for a dataset with a large number of classes, such as ImageNet-21K-P.

## 3.3 Semantic Softmax Training Scheme

Our goal is to develop a dedicated training scheme that utilizes the advantages of both the single-label and the multi-label training. Specifically, our scheme should present for each input image all the available semantic labels, but use softmax activations instead of independent sigmoids to avoid extreme multi-tasking. We also want to have fully accurate ground-truth and training metrics, and provide the network direct data on the semantic hierarchies (this is not achieved even in multi-label training, the hierarchical structure there is implicit). In addition, the scheme should remain efficient in terms of training times.

**Semantic softmax formulation**    To meet these goals, we develop a new training scheme called *semantic softmax* training. As we saw in section 2.2, each label in ImageNet-21K-P can belong to one of 11 possible hierarchies. By definition, for each hierarchy there can be only one ground-truth label per input image. Hence, instead of single-label training with a single softmax, we shall have 11 softmax layers, for the 11 different hierarchies. Each softmax will sample the relevant logits from the

corresponding hierarchy, as shown in Figure 1 (rightmost image). To deal with the partial tagging of ImageNet-21K-P, not all softmax layers will propagate gradients from each sample. Instead, we will activate only softmax layers from the relevant hierarchies. An example is given in Figure 3 - the original image had a label from hierarchy 5. We transform it to 6 semantic ground-truth labels, for hierarchies 0-5, and activate only the 6 first semantic softmax layers (only activated layers will propagate gradients). Compared to single-label and multi-label schemes, semantic softmax training scheme has the following advantages:

1. We avoid extreme multi-tasking ($11, 221$ uncoupled losses in multi-label training). Instead, we have only 11 losses, as the number of softmax layers.
2. We present for each input image all the possible semantic labels. The loss scheme even provides direct data on the hierarchical structure.
3. Unlike single-label and multi-label training, semantic softmax ground-truth and training metrics are fully accurate. If a sample has no labels at hierarchy k, we don't propagate gradients from the kth softmax during training, and ignore that hierarchy for metrics calculation (A dedicated metrics for semantic softmax training is defined in appendix C.3).
4. Calculating several softmax activations instead of a single one has negligible overhead, and in practice training times are similar to single-label training.

**Weighting the different softmax layers** For each input image we have K losses (11). As commonly done in multi-task training [7], we need to aggregate them to a single loss. A naive solution will be to sum them: $L_{\text{tot}} = \sum_{k=0}^{K-1} L_k$ where $L_k$, the loss per softmax layer, is zero when the layer is not activated. However, this formulation ignores the fact that softmax layers at lower hierarchies will be activated much more frequently than softmax layers at higher hierarchies, resulting in over-emphasizing of classes from lower hierarchies. To account for this imbalancing, we propose a balancing logic: let $N_j$ be the total number of classes in hierarchy j (as

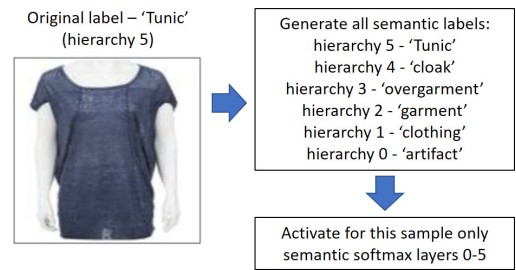

Original label – 'Tunic'
(hierarchy 5)

Generate all semantic labels:
hierarchy 5 - 'Tunic'
hierarchy 4 - 'cloak'
hierarchy 3 - 'overgarment'
hierarchy 2 - 'garment'
hierarchy 1 - 'clothing'
hierarchy 0 - 'artifact'

Activate for this sample only
semantic softmax layers 0-5

**Figure 3: Gradient propagation logic of semantic softmax training.**

presented in Figure 4). Due to the semantic structure, the relative number of occurrences of hierarchy k in the loss function will be: $O_k = \sum_{j=k}^{K-1} N_j$. Hence, to balance the contribution of different hierarchies we can use a normalization factor $W_k = \frac{1}{O_k}$, and obtain a balanced aggregation loss, that will be used for semantic softmax training:

$$L_{\text{tot}} = \sum_{k=0}^{K-1} W_k L_k \tag{2}$$

### 3.4  Semantic Knowledge Distillation

Knowledge distillation (KD) is a known method to improve not only upstream, but also downstream results [58, 59, 61]. We want to combine our semantic softmax scheme with KD training - *semantic KD*. In addition to the general benefit from KD training [19], for ImageNet-21K-P semantic KD has an additional benefit - it can predict the missing tags that arise from the inconsistent tagging. For example, for the left picture in Figure 2, the teacher model can predict the missing labels - 'cow, placental, mammal, vertebrate'. To implement semantic KD loss, for each hierarchy we will calculate both the teacher and the student the corresponding probability distributions $\{T_i\}_{i=0}^{K-1}$, $\{S_i\}_{i=0}^{K-1}$. The KD loss of hierarchy $i$ will be:

$$L_{\text{KD}_i} = \text{KDLoss}(T_i, S_i) \tag{3}$$

where KDLoss is a standard measurement for the distance between distributions, that can be chosen as Kullback-Leibler divergence [19, 58], or as MSE loss [2, 52]. We have found that the latter converges faster, and used it. A vanilla implementation for the total loss will be a simple sum of the losses from different hierarchies: $L_{\text{KD}} = \sum_{i=0}^{K-1} L_{\text{KD}i}$. However, this formulation assumes that all the hierarchies are relevant for each image. This is inaccurate - usually higher hierarchies represent subspecies of animals or plants, and are not applicable for a picture of a chair, for example. So we

need to determine from the teacher predictions which hierarchies are relevant, and weigh the different losses accordingly. Let's assume that for each hierarchy we can calculate the teacher confidence level, $P_i$. A confidence-weighted KD loss will be:

$$L_{\text{KD}} = \sum_{i=0}^{K-1} P_i L_{\text{KD}i} \qquad (4)$$

Eq. 4 is our proposed semantic KD loss. In appendix F we present a method to calculate the teacher confidence level, $P_i$, from the teacher predictions, similar to [58].

## 4 Experimental Study

In this section, we will present upstream and downstream results for the different training schemes, and show that semantic softmax pretraining outperforms single-label and multi-label pretraining. We will also demonstrate how semantic KD further improves results on downstream tasks.

### 4.1 Upstream Results

In appendix C we provide upstream results for the three training schemes. Since each scheme has different training metrics, we cannot use these results to directly compare (pre)training quality.

### 4.2 Downstream Results

To compare the pretrain quality of different training schemes, we will test our models via transfer learning. To ensure that we are not overfitting a specific dataset or task, we chose a wide variety of downstream datasets, from different computer-vision tasks. We also ensured that our downstream datasets represent a variety of domains, and have diverse sizes - from small datasets of thousands of images, to larger datasets with more than a million images. For single-label classification, we transferred our models to ImageNet-1K [30], iNaturalist 2019 [55], CIFAR-100 [29] and Food 251 [25]. For multi-label classification, we transferred our models to MS-COCO [34] and Pascal-VOC [16] datasets. For video action recognition, we transferred our models to Kinetics 200 dataset [26]. In appendix D we provide full training details on all downstream datasets.

**Comparing different pretraining schemes** In Table 2 we compare downstream results for three pretraining schemes: single-label, multi-label and semantic softmax. We see that on 6 out of 7

| Dataset | Single Label Pretrain | Mutli Label Pretrain | Semantic Softmax Pretrain | Semantic Softmax Pretrain + KD |
|---|---|---|---|---|
| ImageNet1K[1] | 81.1 | 81.0 | 81.4 | 82.2 |
| iNaturalist[1] | 71.5 | 71.0 | 72.0 | 72.7 |
| Food 251[1] | 75.4 | 75.2 | 75.8 | 76.1 |
| CIFAR 100[1] | 89.5 | 90.6 | 90.4 | 91.7 |
| MS-COCO[2] | 80.8 | 80.6 | 81.3 | 82.2 |
| Pascal-VOC[2] | 88.1 | 87.9 | 89.7 | 89.8 |
| Kinetics 200[3] | 81.9 | 81.9 | 83.0 | 84.4 |

**Table 2: Comparing downstream results for different pretraining schemes.** Darker cell color means better score. Dataset types and metrics: (1) - single-label, top-1 Acc.[%] ; (2) - multi-label, mAP [%]; (3) - action recognition, top-1 Acc. [%].

datasets tested, semantic softmax pretraining outperforms both single-label and multi-label pretraining. In addition, we see from Table 2 that single-label pretraining performs better than multi-label pretraining (scores are higher on 5 out of 7 datasets tested).

These results support our analysis of the pros and cons of the different pretraining schemes from Section 3: with multi-label training, we have more information per input image, but the optimization process is less efficient due to extreme multi-tasking and extreme imbalancing. All-in-all, multi-label training does not improve downstream results. Single-label training, despite its shortcomings from

the partial tagging methodology and the minimal information per image, provides a better pretraining baseline. Semantic softmax scheme, which utilizes semantic data without the optimization pitfalls of extreme multi-label training, outperforms both single-label and multi-label training.

**Semantic KD**  In Table 2 we also compare the downstream results of semantic softmax pretraining, with and without semantic KD. We see that on all tasks and datasets tested, adding semantic KD to our pretraining process improves downstream results. Indeed the ability of semantic KD to fill in the missing tags and provide a smoother and more informative ground-truth is translated to better downstream results. In appendix G we compare single-label pretraining with KD, to semantic softmax pretraining with semantic KD, and show that the latter achieves better results on downstream tasks.

## 5 Results

In the previous chapters we developed a dedicated pretraining scheme for ImageNet-21K-P dataset, semantic softmax, and showed that it outperforms two baseline pretraining schemes, single-label and multi-label, in terms of downstream results. Now we wish to compare our semantic softmax pretraining on ImageNet-21K-P to other known pretraining schemes and pretraining datasets.

### 5.1 Comparison to Other ImageNet-21K Pretraining Schemes

We want to compare our proposed training scheme to other ImageNet-21K training schemes from the literature. However, to the best of our knowledge, no previous works have published their upstream results on ImageNet-21K, or shared thorough details about their training scheme or preprocessing stage. Recently, prominent new models called ViT [14] and Mixer [53] were published, and official pretrained weights were released [18]. In Table 3 we compare downstream results when using the official ImageNet-21K weights, and when using weights from semantic softmax pretraining.

| | ViT-B-16 | | Mixer-B-16 | |
|---|---|---|---|---|
| Dataset | Official ImageNet-21K Pretrain | Our ImageNet-21K Pretrain | Official ImageNet-21K Pretrain | Our ImageNet-21K Pretrain |
| ImageNet1K[1] | 83.3 | **83.9** | 79.7 | **82.0** |
| iNaturalist[1] | 71.7 | **73.1** | 62.2 | **66.6** |
| Food 251[1] | 74.6 | **76.0** | 69.9 | **74.5** |
| CIFAR 100[1] | 92.7 | **94.2** | 85.5 | **92.3** |
| MS-COCO[2] | 81.1 | **82.6** | 74.1 | **80.9** |
| Pascal-VOC[2] | 78.7 | **93.1** | 63.1 | **88.6** |
| Kinetics 200[3] | 82.7 | **84.1** | 79.3 | **82.1** |

Table 3: **Comparing downstream results for different pretraining schemes.** Dataset types and metrics: (1) - single-label, top-1 Acc.[%] ; (2) - multi-label, mAP [%]; (3) - action recognition, top-1 Acc. [%].

| Dataset | MobileNetV3 | | OFA595 | | ResNet50 | | TResNet-M | | TResNet-L | |
|---|---|---|---|---|---|---|---|---|---|---|
| | 1K | 21K | 1K | 21K | 1K | 21K | 1K | 21K | 1K | 21K |
| iNaturalist[1] | 62.4 | **65.0** | 69.0 | **71.5** | 66.8 | **71.4** | 70.1 | **72.7** | 72.4 | **74.8** |
| CIFAR100[1] | 86.7 | **88.5** | 88.3 | **90.3** | 86.8 | **90.3** | 89.5 | **91.7** | 90.2 | **92.5** |
| Food 251[1] | 70.1 | **70.3** | 72.9 | **73.5** | 72.2 | **74.0** | 75.1 | **76.1** | 76.3 | **77.0** |
| MS-COCO[2] | 73.0 | **74.9** | 74.9 | **77.7** | 76.7 | **80.5** | 79.5 | **82.2** | 81.1 | **83.7** |
| Pascal-VOC[2] | 72.1 | **72.4** | 72.4 | **81.5** | 86.9 | **87.9** | 85.8 | **89.8** | 88.2 | **92.5** |
| Kinetics200[3] | 72.2 | **74.3** | 73.2 | **78.1** | 78.2 | **81.3** | 80.5 | **84.3** | 82.1 | **84.6** |

Table 4: **Comparing downstream results for ImageNet-1K standard pretraining, and our proposed ImageNet-21K-P pretraining scheme.** (1) - single-label dataset, top-1 Acc [%] metric; (2) - multi-label dataset, mAP [%] metric; (3) - action recognition dataset, top-1 Acc [%] metric.

We see from Table 3 that our pretraining scheme significantly outperforms the official pretrain, on all downstream tasks tested. Previous works have observed that MLP-based models can be harder and less stable to use in transfer learning since they don't have inherent translation inductive bias

[6, 39, 35]. When using the official weights, we also noticed this phenomenon on some datasets (Pascal-VOC, for example). Using semantic softmax pretraining, the transfer learning training was more stable and robust, and reached higher accuracy.

## 5.2 Comparison to ImageNet-1K Pretraining

In Table 4 we compare downstream results, for different models, when using ImageNet-1K pretraining (taken from [57]), and when using our ImageNet-21K-P pertraining. We can see that our pretraining scheme significantly outperforms standard ImageNet-1K pretraining on all datasets, for all models tested. For example, on iNaturalist dataset we improve the average top-1 accuracy by 2.9%.

Notice that some previous works stated that pretraining on a large dataset benefits only large models [27, 49]. MobileNetV3 backbone, for example, has only 4.2M parameters, while ViT-B model has 85.6M parameters. Previous works assumed that a large number of parameters, like ViT has, is needed to properly utilize pretraining on large datasets. However, we show consistently and significantly that even small mobile-oriented models, like MobileNetV3 and OFA-595, can benefit from pretraining on a large (publicly available) dataset like ImageNet-21K-P. Due to their fast inference times and reduced heating, mobile-oriented models are used frequently for deployment. Hence, improving their downstream results by using better pretrain weights can enhance real-world products, without increasing training complexity or inference times.

## 5.3 ImageNet-1K SoTA Results

In Table 10 in appendix H we bring downstream results on ImageNet-1K for different models, when using ImageNet-21K-P semantic softmax pretraining. To achieve top results, similar to previous works [33, 61, 58], we added standard knowledge distillation loss into our ImageNet-1K training. To the best of our knowledge, for all the models in Table 10 we achieve a new SoTA record (for input resolution 224). Unlike previous top works, which used private datasets [49], we are using a publicly available dataset for pretraining. Note that the gap from the original reported accuracies is significant. For example, MobileNetV3 reported accuracy was 75.2% [21] - we achieved 78.0%; ResNet50 reported accuracy was 76.0% [20] - we achieved 82.0%.

## 5.4 Additional Comparisons and Results

In appendix J we bring additional comparisons: (1) Comparison to Open Images pretraining; (2) Downstream results comparison on additional non-classification computer-vision tasks; (3) Impact of different number of training samples on upstream results.

## 6 Conclusion

In this paper, we presented an end-to-end scheme for high-quality efficient pretraining on ImageNet-21K dataset. We start by standardizing the dataset preprocessing stage. Then we show how we can transform ImageNet-21K dataset into a multi-label one, using WordNet semantics. Via extensive tests on downstream tasks, we demonstrate how single-label training outperforms multi-label training, despite having less information per image. We then develop a new training scheme, called semantic softmax, which utilizes ImageNet-21K hierarchical structure to outperform both single-label and multi-label training. We also integrate the semantic softmax scheme into a dedicated knowledge distillation loss to further improve results. On a variety of computer vision datasets and tasks, different architectures significantly and consistently benefit from our pretraining scheme, compared to ImageNet-1K pretraining and previous ImageNet-21K pretraining schemes.

**Broader Impact** In the past, pretraining on ImageNet-21K was out of scope for the common deep learning practitioner. With our proposed pipeline, high-quality efficient pretraining on ImageNet-21K will be more accessible to the deep learning community, enabling researchers to design new architectures and pretrain them to top results, without the need for massive computing resources or large-scale private datasets. In addition, our findings that even small mobile-oriented models significantly benefit from large-scale pretraining can be used to enhance real-world products. Finally, our improved pretraining scheme on ImageNet-21K can support prominent MLP-based models that require large-scale pretraining, like ViT and Mixer.

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
