# OpenReview forum: "ImageNet-21K Pretraining for the Masses"
_NeurIPS.cc/2021/Track/Datasets_and_Benchmarks/Round1 — NeurIPS 2021 Datasets and Benchmarks Track (Round 1)_

### Official Review · Reviewer_xKz5 · 2021-06-28
**A interesting work to reform and improve the usability of ImageNet-21K, more clarifications are essential.**

**Rating:** 6
**Confidence:** 4
**Correctness:** Basically, the evaluation is sound an…
**Clarity:** Well written and easy to follow.

**Strengths:**

+ Reforming the ImageNet-21K really helps the vision community.
+ Hierarchical taxonomy labels and the corresponding scheme are valuable for many vision tasks.
+ The experiments are extensive and show impressive results on some tasks.
+ Interesting results are presented, such as the certain improvements with the reformed dataset and various basic models like MobileNet.

**Weaknesses:**

- Some essential details are missing that can make the experiments and proposed scheme more solid. For example:
1. The logit sampling with the semantic Softmax pertaining.
2.  Sec. 3.4, please give more details about the pipeline, and implementation would be better.
3. L260: MSE converges faster. Some analyses or insights behind this? Especially the relationship with the hierarchies.
4. About the difference of improvements on different datasets like ImageNet1K, Food 251, and MS-COCO. I think this is very important to analyze the proposed dataset.
5. L291-L297: Some deeper analysis about the training period, gradient, etc.?
6. Is it possible to conduct some tests to compare the proposed ImageNet-21K-P and the other large-scale datasets? This would further improve the impact and quality of this work. The authors have honestly mentioned JFT-300M in the supplementary (L686). I appreciate this.



**Additional Feedback:**

Advice but may be impractical. Compared to the experimental contribution, the data contribution is relatively smaller. If possible, maybe more annotations to fully build the hierarchical class tree (finer class labels, e.g., the children of the highest class label) and fix the inconsistent tagging would be better. But this could be costly (this would not degrade the rating).

Overall, this paper is beneficial to the vision community, though more details and clarification would make it better to meet the high-quality requirement of NeurIPS. I look forward to the response from the authors and would adjust the rating according to the discussions.

Post-rebuttal:
Thanks for the response from the authors. My concerns about the method and experiment details are basically addressed. And I appreciate the supplemented discussion of the dataset comparison and new results. The other reviews raise some other concerns, but I think a major revision will qualify this work for this track after revision. Though I still think more hierarchy labels will make more dataset contributions as the goal of this track, I stand on my rating.

**Documentation:**

Details are sufficient. URL is given including code, readme, and pre-trained models.

**Ethics:**

N/A.

**Relation To Prior Work:**

Basically good.
But if there are some discussions about the recent large-scale datasets (object-related datasets) and the downstream tasks would be better.

**Summary And Contributions:**

This paper reforms the ImageNet-21K to clean the object classes resplit the data, and organize the hierarchical class taxonomy labels. Subsequently, the authors propose a semantic softmax scheme to fully utilize the hierarchical labels and avoid the cons of multi-label training. Interesting results are presented on pretraining and transfer learning experiments. Moreover, some details are missing and need to be better clarified. And the experimental contribution is larger than the data contribution.

Overall, this is good work with extensive experiments and can help the community better use the bigger ImagetNet-21K.

---

> ### Author Response · Authors · 2021-07-12
> **Response to Paper20 Reviewer xKz5**
>
> We thank the reviewer for the valuable feedback, and will try to address the issues raised:
> 1) **Regarding logits sampling**:
> The logit sampling logic is a fixed assignment between each class to its corresponding hierarchy. Example of classes from different hierarchies appears in Table 1 in the paper.
> To clarify this issue, we will add to paper a pseudo-code with this logic, similar to our supplementary code already released : https://github.com/Alibaba-MIIL/ImageNet21K/blob/main/src_files/semantic/semantics.py#L51
>
> 2) **Regarding KD logic**:
> Our KD logic is a weighted sum of KD losses from the different hierarchies. To clarify this issue, we will add to the paper a pseudo-code, similar to our supplementary code already released:
> https://github.com/Alibaba-MIIL/ImageNet21K/blob/main/src_files/semantic/semantics.py#L104
>
> 3) **Regarding MSE Vs KL-Loss**:
> Indeed, KL-Loss is more commonly used for Knowledge distillation. However, we observed empirically that MSE-Loss usually gives similar accuracies, but convergences faster. We will add an ablation study to the paper to demonstrate this issue.
>
> 4) **Regarding different improvements for different datasets**:
> - Following suggestions from reviewer 1, we will add comparisons with pretrain more computer-vision tasks, such as object detection and image retrieval, to further demonstrate and validate the improvement in pretrain quality for downstream tasks – see https://github.com/Alibaba-MIIL/ImageNet21K/blob/main/ImageNet21K_P_NIPS_rebuttal_experiments.pdf, Table 1 and Table 2.
>
> - One factor that can impact the level of improvement is the similarity between the upstream and downstream datasets. We can expect that if the upstream and downstream datasets are more “similar” in some sense, the possible improvement will be larger. For smaller downstream datasets, we might also expect that the impact of better pretraining will be larger.
> However, other factors may also impact the possible level of improvement, such as labeling errors and inconsistencies with the downstream dataset.  The type of metric used also affects the possible level of improvement. In addition, we cannot directly compare metrics of single-label training (top-1 accuracy) to metrics of multi-label training (mAP).
>
> - Table 3 and Table 4 in the paper show that our pretraining scheme consistently outperforms other schemes. The actual level of improvement is influenced by the factors we stated above (and others). We will clarify this further in the paper.
>
> **Regarding deeper analysis of L291-L297 (why multi-label is not optimal)**:
> The fact that we have extreme multi-tasking (11K competing tasks) in multi-label training can lead to convergence problems due to gradient interference and different gradient magnitudes. There are methods in the literature that analyze these problem and try to tackle them (PCGRad, GradNorm), but they don’t scale well for large datasets like ImageNet-21K. The large positive-negative imbalance is another factor that hinders multi-label training. We showed in the paper that with ASL loss, which specifically tackles the positive-negative imbalance, this problem is mitigated, as can be seen in Table 6 and Table 7. We will try to further clarify this issue in the paper.
>
> **Regarding comparison to other large-scale datasets**:
> Following your suggestion, we will add a comparison to another large-scale dataset, Open Images, a multi-label dataset containing 6M images with more than 9,000 classes.
> Comparison of downstream results appears in file https://github.com/Alibaba-MIIL/ImageNet21K/blob/main/ImageNet21K_P_NIPS_rebuttal_experiments.pdf, Table 4. As we can see, ImageNet21K pretraining consistently provides better downstream results than Open Images dataset. A possible reason is that Open Images, as a multi-label dataset with many classes, suffers from the same optimization pitfalls we described in the paper. We will add this interesting comparison to the paper.

---

> > ### Comment · Reviewer_xKz5 · 2021-07-12
> > **Response to the author response**
> >
> > Thanks for the response, especially for the supplementary experiment and details.
> > My main concerns listed above are basically addressed. Hope the authors can add these contents to the paper.
> > Also looking forward to more discussions and responses to the other reviewers' questions.

---

### Official Review · Reviewer_fntt · 2021-07-02
**interesting improvement to imagenet-21k**

**Rating:** 7
**Confidence:** 4
**Correctness:** Not much to add besides what I alread…
**Clarity:** Yes, it is well written.

**Strengths:**

* A practically relevant dataset that I expect to be useful to most researchers doing computer vision & deep learning
* A thorough set of ablations that show the significance of the data vs losses used vs model vs downstream tasks

**Weaknesses:**

* I am not convinced by the need to have the images resized the way they were resized
* The semantic/hierarchical softmax loss used in the work is a bit niche, since it's custom for this dataset and I'm not sure how generalizeable it is to anything outside of it.

**Additional Feedback:**

N/A

**Documentation:**

Yes, it's well documented.

**Ethics:**

Nothing really that stood out to me.

**Relation To Prior Work:**

yes.

**Summary And Contributions:**

This work modifies and improves the ImageNet-21k datasets by:

* preprocessing it
* using WordNet to do semantic multi-labeling of each image
* developing a semantic softmax scheme (not technically an improvement to the dataset I suppose)

In general, this is a nice piece of work. Here are some comments:
* The authors say that "classes with few samples cannot be learned efficiently". Is this true? There's a whole lot of research out there about few-shot learning with as few as 5 or 10 labels. Limiting the dataset to 500 labels per class or more seems ... limiting.
* I think the value of doing the squish-resizing -- having the dataset be smaller -- is not that big compared to the potential loss of various data augmentation techniques that one could do on the original images. ˆ Is there a version of the data that is not resized?
* I don't necessarily buy the argument on line 186 that multi-label training suffers from more local mimima than single softmax training. Has this been shown empirically (or otherwise) by the authors?
* The semantic softmax loss in Section 3.3 seems very tailored to the dataset itself. It's not obvious to me that any of the pieces from it are really usable outside the narrow confines of the hierarchy implicit in this dataset.
* The pretraining experiments seem thorough and the ablations are very detailed. It's clear that this dataset has practical merits so I expect that others will end up actually using it.

---

> ### Author Response · Authors · 2021-07-12
> **Response to Paper20 Reviewer fntt**
>
> We appreciate the positive review! We will try to address the issues raised:
>
> **Regarding removal of classes with few samples**:
> Please note that removing classes with less than X samples was used in earlier releases of ImageNet [1]. It is also a common practice when processing other large-scale datasets, like Open Images [2] and LVIS [3]. There are indeed works on few-shot learning with as few as 5 or 10 labels, but since our focus was to produce an overall better pretraining, we chose to avoid rare classes, that may add complications to the optimization process. Following your suggestion, we will add a dedicated ablation to the paper to demonstrate that removing rare classes indeed contributes to pretrain quality.
>
> **Regarding squish-resizing**:
> The official ImageNet site (Image-net.org) offers two versions of ImageNet21K to download: the original (unresized) dataset, and our processed squish-resized version.
> We also shared the script that was used to process the original dataset:
> https://github.com/Alibaba-MIIL/ImageNet21K/blob/main/dataset_preprocessing/processing_script.sh. Following your suggestion, we will share another version of the processing script, that will do short-edge resizing instead of squish resizing.
>
> **Regarding optimization issues in multi-label pretraining**:
> We show empirically in the paper that the ASL loss, which directly aims at mitigating the positive-negative imbalance in multi-label classification, leads to a significantly better pretraining quality compared to the regular CE loss (Table 7). This supports our claim of positive-negative imbalance as a possible optimization pitfall of multi-label training.
> \
> Following a suggestion from reviewer 3, we will also add to the paper a comparison to another large-scale dataset that can be used for pretraining, Open Images. This is a multi-label dataset, with 6M images and more than 9000 classes. Example results can be seen in Table 4 in:
> https://github.com/Alibaba-MIIL/ImageNet21K/blob/main/ImageNet21K_P_NIPS_rebuttal_experiments.pdf.
> This comparison shows that ImageNet-21K semantic softmax training achieves better pretrain quality, and further hints that possible optimization problems in multi-label training (for example, extreme multi-tasking) lead to lower pretrain quality.
> \
> \
> [1] Deng et al, What does classifying more than 10,000 image categories tell us?
>
> [2] Durand et al, Learning a deep convnet for multi-label classification with partial labels
>
> [3] Kundu et al, Exploiting weakly supervised visual patterns to learn from partial annotations

---

### Official Review · Reviewer_wWou · 2021-07-03
**While paper can be found useful to a group of developers and researchers (especially those interested in participating in CV challenges where squeezing accuracy even for 1-2% is critical), the contributions don’t seem sufficient in my opinion for NeurIPS. The paper is an incremental work and the contribution and scope might be best fit for a related workshop.**

**Rating:** 4
**Confidence:** 3
**Correctness:** The claims seem to be correct.
**Clarity:** The paper is well written.

**Strengths:**

The paper shows how full imageNet dataset with over 14M images and 21K classes can be used to train better backbones as part of the pre-training steps for few supervised tasks. The paper claims that the contributions are mainly around cleaning up the dataset, providing official test/train/val splits, Introducing efficient training techniques and providing results on down-stream tasks to show effectiveness of the newly trained backbone. (Side note: The final cleaned dataset has only 11K categories. So probably the name ImageNet21K is a bit confusing when compared to ImageNet-1K.)

**Weaknesses:**

While paper can be found useful to a group of developers and researchers (especially those interested in participating in CV challenges where squeezing accuracy even for 1-2% is critical), the contributions don’t seem sufficient in my opinion for NeurIPS. The paper is an incremental work and the contribution and scope might be best fit for a related workshop. The steps around cleaning the dataset is pretty standard. Training schemes used for pre-trained models are interesting, while some are well known and not novel (e.g Knowledge Distillation) the other one like semantic softmax seems to have some novelty. However, I would still like to see a more thorough literature review and how this approach compare to wide range of work on extreme classification tasks or classification tasks with noisy data (http://manikvarma.org/events/XC16/index.html). Breaking up large classification problems to smaller sub-task classification is not new. Additionally the experiment are not complete in my opinion. You can refer to some of my comments below regarding the experiments.

Additionally, one might argue the value of such dataset for the community. Increasing the size of the dataset for pre-training might not necessarily be valuable at least for a big portion of research community as it requires a significantly more time to process and pre-train backbones where the reward can be incremental improvements. The resources for pre-training such models might not be available to everyone and smaller dataset might indeed be desirable. This specifically can become a bottleneck when trying new backbones where pre-trained models don’t exist.


Experiments:

1) One experiment that would be interesting and I wish it was included in the paper is to see is how much adding data actually helps. Do we really need all the 14M and 11K classes to achieve some of the results reported in the paper? For example, if we generate ImageNet1K, 2K, 5K, 10K and repeat the experiments, do we at some  point see that the results plateau.

2) Majority of the experiments are still around classification. Either in videos or images. Single label and multi-label. These tasks are more likely to benefit from a stronger backbone training for similar tasks on larger images as it can better generalizes. It would be interesting to see how such backbone can increase the quality of other popular CV tasks, like detection, segmentation and etc.

3)The improvement from official 21K pre-training to the proposed method in this paper is usually within 1-2%. However, the there is one dataset where the improvements are close to 15% (Pascal-VOC, Table 3). Would be useful if authors provide some intuition on why the improvement is much larger in this dataset.

4)Pre-training used to be an important part of classical computer vision tasks such as detection. Mainly for faster convergence as well as small size of dataset trained for target task. However, the size of the dataset and labels are increasing every year and the impact of pre-trianing may diminish. It would be interesting to see an albation study of such experiments in this paper.

5)The numbers reported in the paper for training time time, which in my opinion is an important aspect is shown at over 2days on 8x V100 GPUs. However, this is for when models are initialized form ImageNet1K training which is a bit counter intuitive (L:148 We initialize our model from ImageNet-1K training). I would be curious to see how much longer the training would be if ImageNet1k is not used. It is counter intuitive if one needs to include 3 rounds of training for each backbone (ImageNet1K -> ImageNet21K -> Downstream Task)

6)ImageNet-21K dataset labels are still noisy. One may argue that instead of the semantic softmax training or multi-label training one could stick with single-label training and use some higher level of hierarchy where the #labels can be reduced drastically. Would be interesting to see such experiments as well


**Additional Feedback:**

I have added some comments on the experiments section and how that can be improved:

Experiments:

1) One experiment that would be interesting and I wish it was included in the paper is to see is how much adding data actually helps. Do we really need all the 14M and 11K classes to achieve some of the results reported in the paper? For example, if we generate ImageNet1K, 2K, 5K, 10K and repeat the experiments, do we at some point see that the results plateau.

2) Majority of the experiments are still around classification. Either in videos or images. Single label and multi-label. These tasks are more likely to benefit from a stronger backbone training for similar tasks on larger images as it can better generalizes. It would be interesting to see how such backbone can increase the quality of other popular CV tasks, like detection, segmentation and etc.

3)The improvement from official 21K pre-training to the proposed method in this paper is usually within 1-2%. However, the there is one dataset where the improvements are close to 15% (Pascal-VOC, Table 3). Would be useful if authors provide some intuition on why the improvement is much larger in this dataset.

4)Pre-training used to be an important part of classical computer vision tasks such as detection. Mainly for faster convergence as well as small size of dataset trained for target task. However, the size of the dataset and labels are increasing every year and the impact of pre-trianing may diminish. It would be interesting to see an albation study of such experiments in this paper.

5)The numbers reported in the paper for training time time, which in my opinion is an important aspect is shown at over 2days on 8x V100 GPUs. However, this is for when models are initialized form ImageNet1K training which is a bit counter intuitive (L:148 We initialize our model from ImageNet-1K training). I would be curious to see how much longer the training would be if ImageNet1k is not used. It is counter intuitive if one needs to include 3 rounds of training for each backbone (ImageNet1K -> ImageNet21K -> Downstream Task)

6)ImageNet-21K dataset labels are still noisy. One may argue that instead of the semantic softmax training or multi-label training one could stick with single-label training and use some higher level of hierarchy where the #labels can be reduced drastically. Would be interesting to see such experiments as well

-------------------------------------------

Post Rebuttal: I appreciate the authors response. The additional experiments and ablation studies are definitely adding more isights. However, my concerns regarding the contribution of the paper and it's novelties are still in place. The work is incremental and doesn't include enough novelty.

I don't agree with some of the comments on value of the dataset authors provided in rebuttal. There are many datasets out there for pre-training and one can't just publish a paper by using a bigger dataset to pre-train their model and show better results compare to SOTA (as there is no novelty there). So the adaption of a bigger dataset for pre-training for any paper will still require all the baselines and previous SOTAs to also be trained with the new pre-training dataset. This consequently requires a lot of computing resources and I'm surprised authors didn't provide numbers on the training time without using ImageNet 1K as pre-training step. This is the easiest experiments that authors could have conducted in the rebuttal, unless it takes a long time to train/converge such model without pre-training step that didn't fit in the 1 week rebuttal period.

I would like to stand with my original rating.

**Documentation:**

The documentation is sufficient.

**Ethics:**

The authors are not introducing new dataset as the work is using the already existing ImageNet dataset.

**Relation To Prior Work:**

Please refer to my comments regarding the experiments and contributions. There are areas that I would like to see more comparison especially around semantic softmax compare with prior work dealing with large number of classes and noisy labels as deviding a large scale classification task into smaller sub-problems is not new.

**Summary And Contributions:**

The paper shows how full imageNet dataset with over 14M images and 21K classes can be used to train better backbones as part of the pre-training steps for few supervised tasks. The paper claims that the contributions are mainly around cleaning up the dataset, providing official test/train/val splits, Introducing efficient training techniques and providing results on down-stream tasks to show effectiveness of the newly trained backbone. (Side note: The final cleaned dataset has only 11K categories. So probably the name ImageNet21K is a bit confusing when compared to ImageNet-1K.)

---

> ### Author Response · Authors · 2021-07-12
> **Response to Paper20 Reviewer wWou**
>
> We thank the reviewer for the insightful experiments suggested, and the thorough review. We will try to address the issues raised.
>
> **Regarding the experiments suggested**:
> 1) Testing how much data actually helps:
> We followed the experiment suggested, i.e., generated ImageNet1K, 2K, 5K, 10K variants, in addition to the original dataset (11.7K). Results appear in
> https://github.com/Alibaba-MIIL/ImageNet21K/blob/main/ImageNet21K_P_NIPS_rebuttal_experiments.pdf , Figure 1 (upstream results) and Figure 2 (downstream results). As we can see, more training pictures lead to better upstream and downstream scores. We will add this interesting result to the paper.
> 2) Adding extra tasks for pretraining comparison:
> Following your suggestion, we conducted experiments for two more (non-classification) CV tasks: object detection and image retrieval. Results appear in https://github.com/Alibaba-MIIL/ImageNet21K/blob/main/ImageNet21K_P_NIPS_rebuttal_experiments.pdf , Table 1 and Table 2. As we can see, using our pretraining scheme leads to a meaningful improvement also on those non-classification tasks. We will add these results to the paper.
> 3) Regarding the larger gap in accuracies on PASCAL-VOC (and also CIFAR-100, for Mixer model):
> The larger gap stems from the fact that transformers models can be unstable and sensitive to hyper-parameters selection. This was also observed in previous works, for example [1,2]. Our proposed pretraining scheme leads to more stable and robust models, compared to the original pretraining. We offer a full reproduction code, on a public package (timm), that shows this phenomenon:
> https://github.com/Alibaba-MIIL/ImageNet21K/blob/main/Transfer_learning.md.
> The fact that our pretraining scheme can make transformers models more stable and robust is another advantage compared to standard 21K pretraining. We will clarify and further emphasize this issue in the paper.
> 4) Regarding the need for pretraining for large downstream datasets:
> We believe that pretraining is highly beneficial also for large datasets. It enables to significantly reduce training times, and leads to  higher scores.
> To demonstrate this issue, we conducted an experiment on Open Images as a downstream dataset, which contains 6M images labeled with more than 9000 classes. Results appear in: https://github.com/Alibaba-MIIL/ImageNet21K/blob/main/ImageNet21K_P_NIPS_rebuttal_experiments.pdf, Table 3. As we can see, there is a large gap in downstream scores with and without pretraining (6% mAP). This gap might be reduced if we do more epochs, but this will require allocation of more computational resources.
> 5) Regarding initialization from ImageNet1K:
> Indeed, it is possible to train on ImageNet21K from scratch, and reach the same results as initialization from ImageNet1K, but it will require more training epochs. Per your suggestion, we will add this comparison to the paper.
> 6) Regarding using only hierarchy 0 to reduce label noise:
> With semantic softmax scheme, we completely remove the noise from ImageNet21K labels, while still utilizing all the available semantic labels. If we use labels from hierarchy 0 only, we will indeed have no noise, but at the cost of reducing the number of classes drastically, and classifying only to general abstract classes (e.g. people, animal, plant,…). This can be seen as a private case of semantic softmax, with zero weight to hierarchies higher than 0.
>
> **Regarding the contribution of the paper**:
> We developed a novel pretraining scheme for ImageNet-21k, which gives consistent improvement of 1-2% on downstream datasets, over a strong baseline, for multiple models and tasks, without increasing the inference time or downstream training complexity. We believe this is a meaningful contribution.
> Since our proposed scheme is also relevant for mobile-oriented models, it can enhance real-world products, in addition to CV datasets and challenges. We also show that our pretraining scheme stabilizes transformers models, and makes them more robust to hyperparameter selection, which is another meaningful improvement.
>
> **Regarding the value of large dataset to the community**:
> Without large-scale pretraining, there is no practical ability to reach even near-SOTA results on some CV tasks. Our work tries to address this problem. For researchers without access to enough GPU resources, we provide a variety of high-quality pretrained models for downstream tasks. Our proposed pretraining scheme is as efficient and economical as possible, to enable researchers with limited GPU resources to try and pretrain themselves. In addition to the paper, we share our processed dataset, a full reproduction code, pretrained models and more, to maximize the added value for the community.
> \
> \
> [1] Chen et al, An empirical study of training self-supervised visual transformers
> \
> [2] Mosbach et al, On the Stability of Fine-tuning BERT: Misconceptions, Explanations, and Strong Baselines

---

### Decision · Program_Chairs · 2021-07-27

**Decision:**

Accept

**Comment:**

This paper received mixed reviews, leaning positive. The negative review is mainly concerned about the significance of the work. AC believes that the authors' responses have sufficiently addressed these concerns. Overall, AC believes that the paper will be a useful contribution, given that many downstream tasks can benefit.